# OpenReview forum: "Offline RL with No OOD Actions: In-Sample Learning via Implicit Value Regularization"
_ICLR.cc/2023/Conference — ICLR 2023 notable top 5%_

### Official Review · Reviewer_HMTv · 2022-10-23

**Confidence:** 4
**Correctness:** 3
**Technical Novelty And Significance:** 3
**Empirical Novelty And Significance:** 2
**Recommendation:** 8

**Clarity, Quality, Novelty And Reproducibility:**

Clarity: The paper is clear and easy to read.

Quality: Fair. The paper could do a better job of demonstrating the benefits of SQL over IQL.

Originality: Fair. The theoretical results are novel and interesting, however the empirical method is highly motivated by the in-sample Q-learning (IQL) method.

Reproducibility: There are not enough details in the appendix to easily replicate the results on D4RL -- This possibly can be fixed if the code is open-sourced.

**Strength And Weaknesses:**

Strengths:

- The paper is well written, especially the theoretical section.
- The proposed SQL method is theoretically well-motivated. The theory also points out interesting connections to existing methods including CQL and IQL. Furthermore, SQL only requires one hyperparameter as opposed to two in IQL.
- The implicit value framework derived in the framework seems like a general framework for instantiating in-sample Q-learning methods based on $\alpha$-divergences (although most methods might turn out to be intractable).

Weaknesses

- The improvements on the D4RL benchmark do not seem to be statistically significant -- Reported results are likely within 1 standard deviation of baseline results and thus the SOTA claim may not be valid.
   - Error bars are not reported for any of the baselines in Table 1 and 2 -- however, the standard deviation is known to be large for D4RL datasets. See https://openreview.net/pdf?id=Y4cs1Z3HnqL for a paper that reports standard deviation for some of the baseline methods.
   -  The proposed method seems more complex to implement than existing methods such as IQL, while resulting only in marginal gains.
-  Possibly Unfair Comparisons: It seems that per-task hyperparameter tuning is done for SQL while the baseline methods' results seem to be copied from prior papers which used the same hyperparameter for a given domain (Kitchen / Antmaze / MuJoCo).
- It's not clear whether "sparsity" plays an important role in performance of SQL -- this is not explored much in the paper despite the title.
-


**Summary Of The Paper:**

This paper theoretically derives an in-sample offline RL algorithm, Sparse Q-Learning (SQL), using a behavior regularized MDP for offline reinforcement learning. SQL has interesting connections to existing offline RL algorithms including CQL and the in-sample IQL algorithm. SQL performs similarly to prior methods on MuJoCo and Antmaze tasks and better on Kitchen tasks in D4RL. SQL is shown to be superior on custom smaller datasets than CQL as well as more stable than IQL on mixture datasets containing expert and random data.

**Summary Of The Review:**

Overall, the paper presents theoretically well-motivated method for addressing the ad hoc choices in IQL. However, the method doesn't seem to offer much empirical gains over IQL on the standard benchmarks. However, other than robustness results on mixture datasets (noisy data regime in Section 5.1), it is unclear whether SQL offers any benefits over IQL in terms of practical benefits.

Questions / Suggestions:
- Can you provide empirical evidence for whether SQL provide any benefits over IQL in the small-data regime in Section 5.2?
- In-line with the best practices for evaluation, I'd recommend the use of [rliable](https://github.com/google-research/rliable) library [1] to validate the claim for SOTA results. Furthermore, standard deviation be reported for all methods in Table 1.
- Does the method scale to high-dimensional image-based datasets such as the Atari datasets in RL Unplugged [2]?
- Kumar et. al (2022) showed degradation in CQL performance with prolonged training (including on the Antmaze datasets) -- would SQL  provide any benefit compared over CQL / IQL due to being a principled approach for in-sample learning?
- Are there any guidelines to set the $\alpha$ parameter -- given that SQL is theoretically motivated, it would be nice if the hyperparameter tuning is easier / more intuitive than existing methods like CQL / IQL.

Minor:
- Can you clarify how Jensen's inequality is applied in Equation 2?
- The generality of the IVR framework would be seen if another value of $\alpha$ was instantiated to derive in-sample version of another existing method or deriving a new method altogether.


References:

[1] Agarwal, R., Schwarzer, M., Castro, P. S., Courville, A. C., & Bellemare, M. (2021). Deep reinforcement learning at the edge of the statistical precipice. Advances in neural information processing systems, 34, 29304-29320.

[2] Gulcehre, C., Wang, Z., Novikov, A., Paine, T., Gómez, S., Zolna, K., ... & de Freitas, N. (2020). Rl unplugged: A suite of benchmarks for offline reinforcement learning. Advances in Neural Information Processing Systems, 33, 7248-7259.

[3] DR3: Value-Based Deep Reinforcement Learning Requires Explicit Regularization (2022). ICLR. https://openreview.net/forum?id=POvMvLi91f


-------
----------  Post rebuttal ----------

Based on authors's response as well as the updated paper, I am now in favor of acceptance and updated the score to 8 (accept) from 5 (below accept).

---

> ### Author Response · Authors · 2022-11-13
> **Response to Reviewer HMTv (1/2)**
>
> We would like to thank the reviewer for their in-depth review of our manuscript.
>
> >"Can you provide empirical evidence for whether SQL provide any benefits over IQL in the small-data regime in Section 5.2?"
>
> We have updated Table 2 and added the results of IQL, it is shown that both SQL and IQL outperform CQL while SQL achieves better results than IQL.
>
> >"In-line with the best practices for evaluation, I'd recommend the use of rliable library [1] to validate the claim for SOTA results. Furthermore, standard deviation be reported for all methods in Table 1."
>
> Thank you for your suggestions, we have added the standard deviation of two strongest baselines, IQL and CQL. We also use rliable library to validate the claim for SOTA results of SQL, please see Figure 7 for details.
>
> >"Does the method scale to high-dimensional image-based datasets such as the Atari datasets in RL Unplugged [2]?"
>
> Thanks for pointing out that! We think SQL could scale to those datasets given that CQL had achieved nice results on these datasets and SQL is the "in-sample" version of CQL, which owns more training stability. We will add some experiments about this on the latter version.
>
> >"Kumar et. al (2022) showed degradation in CQL performance with prolonged training (including on the Antmaze datasets) -- would SQL provide any benefit compared over CQL / IQL due to being a principled approach for in-sample learning?"
>
> Thanks for pointing out that! We think both IQL and SQL will provide benefit compared over CQL about the training performance degradation. In the DR3 paper, the authors found that large feature dot products arise when *out-of-sample* actions are used in TD-learning compared to SARSA, despite similar Q-values. Note that both IQL and SQL are SARSA-style learning methods, i.e., using only *in-sample* actions.
>
> >"Are there any guidelines to set the parameter -- given that SQL is theoretically motivated, it would be nice if the hyperparameter tuning is easier / more intuitive than existing methods like CQL / IQL."
>
> To give an intuitive guideline of hyperparameter tuning in SQL, we show the relationship of the normalized score and non-sparsity ratio (i.e., $\mathbb{E}_{(s, a) \sim \mathcal{D}}[1 (1+ (Q(s, a) - V(s))/2\alpha >0 )]$) with different $\alpha$ in SQL, in Table 5 and Table 6.
>
> We find the value of non-sparsity ratio is controlled by the hyperparameter $\alpha$, typically a larger $\alpha$ gives less sparsity, sparsity plays an important role in the performance of SQL and we need to choose a proper sparsity ratio to achieve the best performance. The best sparsity ratio depends on the composition of the dataset, for example, the best sparsity ratios in MuJoCo datasets (around 0.1) are always larger than those in AntMaze datasets (around 0.4), this is because AntMaze datasets are kind of multi-task datasets (the start and goal location are different from the current ones), there is a large portion of useless transitions contained so it is reasonable to give those transitions zero weights by using more sparsity.
>
> So the practical guideline of hyperparameter tuning in SQL is that, if the datasets are more diverse and contain a large portion of useless transitions, we should use a lower $\alpha$ to increase the sparsity, otherwise if the datasets are less diverse or require more behavior cloning, we should use a higher $\alpha$ to decrease the sparsity.
>
> >"Can you clarify how Jensen's inequality is applied in Equation 2?"
>
> The regularization term $\mathbb{E}_{\pi} [f(\frac{\pi}{\mu})]$, is equivalent to
>
> $$
> \mathbb{E}_\pi \left[f \left(\frac{\pi}{\mu} \right) \right] = \mathbb{E}_\mu \left[\frac{\pi}{\mu}f \left(\frac{\pi}{\mu} \right) \right]
> $$
>
> because $h_f(x) = x f(x)$ is strictly convex, we can apply Jensen's inequality by moving the expectation inside, which has
>
> $$
> \mathbb{E}_\mu \left[\frac{\pi}{\mu}f \left(\frac{\pi}{\mu} \right) \right] = \mathbb{E}_\mu \left[h_f \left(\frac{\pi}{\mu} \right) \right] \geq h_f \left(\mathbb{E}_\mu \left[\frac{\pi}{\mu} \right] \right) = h_f(1) = 1 f(1) = 0
> $$
>
> >"The generality of the IVR framework would be seen if another value of  was instantiated to derive in-sample version of another existing method or deriving a new method altogether."
>
> Thanks for pointing out that! We derive another practical new offline RL algorithm, EQL, by applying the reverse KL divergence, we find EQL is an "implicit" version of AWR/AWAC that avoids any out-of-distribution action.
>
> We also test EQL on D4RL benchmark datasets and noisy datasets used in our experiments. It is shown that EQL also has a strong empirical performance. Please see Appendix A for details.
>
> >"...although most methods might turn out to be intractable)."
>
> We discuss about the feasibility of applying other valid choices of $\alpha$-divergence. Please see Appendix A for details.

---

> > ### Author Response · Authors · 2022-11-13
> > **Response to Reviewer HMTv (2/2)**
> >
> > >"It seems that per-task hyperparameter tuning is done for SQL while the baseline methods' results seem to be copied from prior papers which used the same hyperparameter for a given domain"
> >
> > Thanks for your suggestions, for a fair comparision, we re-run IQL on all datasets and report the score of IQL by choosing the best score from $\tau$ in $[0.5, 0.6, 0.7, 0.8, 0.9, 0.99]$, using author-provided implementation (https://github.com/ikostrikov/implicit_q_learning). We also re-run CQL on all datasets and report the best score from $\texttt{min-q-weight}$ in $[0.5, 1, 2, 5, 10]$, using a popular PyTorch-version implementation (https://github.com/young-geng/CQL).
> >
> > We have updated the scores in Table 1 in our paper, we find that the reported scores in CQL and IQL papers are almost the highest one over the searched hyperparameters, SQL still outperforms both CQL and IQL, please refer to the updated paper for details.
> >
> > >"It's not clear whether "sparsity" plays an important role in performance of SQL "
> >
> > Thanks for pointing out that. The sparsity actually helps to learn a better value function (as we initially claimed in Section 4.4), we have added a whole section to elaborate how does sparsity benefit value learning in SQL, in both tabular setting and continuous action setting, please refer to Appendix B for details.
> >
> > >"however the empirical method is highly motivated by the in-sample Q-learning (IQL) method. "
> >
> > We respectfully disagree with it. Our empirical methods (both SQL and newly added EQL) are derived from the IVR framework, which is obtained by solving a behavior-regularized MDP. Our methods are not motivated by IQL, in fact, we explain why certain choices are used in IQL and also demonstrate its weaknesses.

---

> > > ### Comment · Reviewer_HMTv · 2022-11-13
> > > **Thanks for the response.**
> > >
> > > Dear authors, thank you for your thorough response. I am now in favor of acceptance and will update my score to 6.
> > >
> > > Some minor follow-ups:
> > >
> > > > We have added the standard deviation of two strongest baselines, IQL and CQL. We also use rliable library to validate the claim for SOTA results of SQL, please see Figure 7 for details.
> > >
> > > Can you please also add a description for performance profiles & provide a reference to make the paper self-sufficient.  Also, based on the profiles, I can see percentiles too in addition to aggregate metrics, and it does look like SQL is somewhat better than IQL / CQL.
> > >
> > > > We think SQL could scale to those datasets given that CQL had achieved nice results on these datasets and SQL is the "in-sample" version of CQL, which owns more training stability. We will add some experiments about this on the latter version.
> > >
> > > Indeed, experiment on image-based offline datasets would make the paper significantly stronger. I would be willing to raise my score further if you are able to obtain strong results on such datasets (for example, the commonly used 1%, 5% offline Atari data settings).
> > >
> > > > Derived EQL and benchmarked on D4RL.
> > > This is great -- I think this does show the generality of the proposed framework.

---

> > > > ### Author Response · Authors · 2022-11-19
> > > > **Thank You!**
> > > >
> > > > Thank you very much for engaging with us and for increasing your score! We really appreciate it!
> > > >
> > > > > Can you please also add a description for performance profiles & provide a reference to make the paper self-sufficient.
> > > >
> > > > Thanks for your suggestions! We have added a description in the updated paper.
> > > >
> > > > > Indeed, experiment on image-based offline datasets would make the paper significantly stronger. I would be willing to raise my score further if you are able to obtain strong results on such datasets (for example, the commonly used 1%, 5% offline Atari data settings).
> > > >
> > > > Thanks for your suggestions to make our paper significantly stronger! We have added experiments on image-based offline Atari datasets, we implement discrete SQL (D-SQL) and compare it with D-IQL, D-CQL and D-BCQ on 5% and 10% mixed and medium datasets in 3 games, based on the d3rlpy offline RL library. The aggregated results are shown as follows, SQL still obtains strong results on such datasets.
> > > >
> > > > ---
> > > > -----------We have updated the results by running 5 seeds:-----------
> > > >
> > > > | Task |  D-BCQ |  D-IQL |  D-CQL |  D-SQL |
> > > > |  ----  | ----  |  ----  | ----  | ----  |
> > > > |  breakout-medium-v0 (10%) | 3.5 | 20.2$\pm$7.6 | 11.5$\pm$3.4 | **28.1$\pm$6.3** |
> > > > |  qbert-medium-v0 (10%)  | 395.4 | 3701.4$\pm$454.4 | 3016.4$\pm$415.4 | **4435.4$\pm$440.1** |
> > > > |  seaquest-medium-v0 (10%)  | 438.1 | 315.9$\pm$25.3 | 380.4$\pm$ 29.2| **449.3$\pm$19.1** |
> > > > | breakout-mixed-v0 (10%)  | 8.1 | 10.9$\pm$1.9 | 9.1$\pm$3.7 | **13.6$\pm$2.1** |
> > > > | qbert-mixed-v0 (10%)  | 557.5 | 1055.3$\pm$205.6 | 890.2$\pm$271.2 | **1244.3$\pm$285.3** |
> > > > | seaquest-mixed-v0 (10%)  | 300.3 | **326.8$\pm$42.6** | 306.5$\pm$25.3 | 297.4$\pm$21.7 |
> > > > | breakout-medium-v0 (5%) |  1.6 |  15.1$\pm$4.8 |  8.7$\pm$2.5 |  **16.5$\pm$1.2** |
> > > > | qbert-medium-v0 (5%)  |  301.6 |  2563.3$\pm$383.1 |  **3026.3$\pm$463.2** |  2779.6$\pm$289.2 |
> > > > | seaquest-medium-v0 (5%)  |  301.9 |  324.5$\pm$24.2 |  272.5$\pm$22.3 | **341.3$\pm$24.2** |
> > > > | breakout-mixed-v0 (5%)  |  5.3 |  10.5$\pm$1.1 |  10.8$\pm$0.7 |  **12.3$\pm$1.2** |
> > > > | qbert-mixed-v0 (5%)  |  576.4 |  931.1$\pm$326.1 |  801.3$\pm$228.4 |  **970.7$\pm$155.6** |
> > > > | seaquest-mixed-v0 (5%)  |  275.5 |  292.2$\pm$51.2 |  298.4$\pm$45.2 |  **336.7$\pm$52.6** |
> > > >
> > > > Due to time limit, we are sorry for not being able to compare more baselines on more Atari games, we will add more in the latter revision.

---

> > > > > ### Comment · Reviewer_HMTv · 2022-11-20
> > > > > **Thanks for the additional results.**
> > > > >
> > > > > Thanks for running the additional experiments on the high-dimensional Atari datasets in RL Unplugged. Since error bars / standard deviation is omitted, it is unclear whether D-SQL's performance is within one-standard deviation.
> > > > > Regardless, it seems that D-SQL outperforms BCQ and is comparable or better in performance to both CQL and IQL.
> > > > >
> > > > > Assuming the authors would include results on additional games and at least report results on 2 seeds (min / max scores) or standard deviation on each of the methods (BCQ can be safely skipped), I am increasing my score to an accept.
> > > > >
> > > > > Additional suggestions:
> > > > >
> > > > >  - Re aggregate metrics, can you please report IQM / optimality gap in addition to the performance profile for results in Figure 7 in the final revision in the main paper? Profiles are great but those metrics make it easy to directly compare aggregate performance and are less prone to outliers than mean.

---

> > > > > > ### Author Response · Authors · 2022-11-24
> > > > > > **Thanks Again!**
> > > > > >
> > > > > > Thanks for your additional suggestions and for appreciating our paper by updating the score to 8!
> > > > > >
> > > > > > We have added the standard deviation to the table by running 5 seeds on each algorithm. We promise that we will add IQM / optimality gap in the final revision. We will also include results on additional Atari games and add IQM and the performance profile of results in Atari domains in the final revision.

---

### Official Review · Reviewer_u1bC · 2022-10-24

**Confidence:** 4
**Correctness:** 3
**Technical Novelty And Significance:** 3
**Empirical Novelty And Significance:** 3
**Recommendation:** 8

**Clarity, Quality, Novelty And Reproducibility:**

- Clarity: Overall writing is clear and presentation is good.
- Quality: Overall paper quality is good.
- Novelty: The new method is not that different from existing methods, however, the design of the new method together with the theoretical contributions can be considered novel.
- Reproducibility: Overall the method is not too complex, technical details provided and some discussion on hyperparameters provided.

**Strength And Weaknesses:**

Strengths:
- Overall good writing quality and clarity.
- Interesting theoretical results that might help better understand offline methods that rely on some form of regularization terms.
- Empirical results on D4RL and other 2 comparisons seem to show SQL has stronger performance than alternative methods, and some of these methods are recent methods with good performance.
- Technical details provided for reproducibility.
- a lot of discussion on related works, easy to understand how it relates to prior works

Weaknesses:
- Thank you for providing technical details including your hyperparameter search range. However, I am a bit concerned that it seems you performed a hyperparameter search on each dataset individually, and then select the best-performing one. This makes sense, but for the methods you compare to, are they using the same hyperparameter for all tasks, or do they also search hyperparameter on each individual dataset and report the best one?
- Similarly, in the case when you compare SQL to CQL and IQL on the noisy and small data regime settings, did you also do a thorough hyperparameter search for IQL and CQL?
- Maybe I missed sth but have you provided empirical evidence and figures to demonstrate that your method has a more "sparse" policy than the other competitive methods?
- What is the final hyperparameter you selected for each individual dataset? (You mentioned the best is selected for each dataset, would be good if you also report the actual numbers)
- What about computation efficiency? How fast in wall-clock speed does your method compare to others?
- Hyperparameter sensitivity: for example can you provide some results on how performance changes with different alpha values? This will help other researchers understand how hard it is to tune your method on new tasks.

Not really important:
- some minor typos

Other Questions:
- Will your code be open sourced?

**Summary Of The Paper:**

The paper introduces Implicit Value Regularization (IVR) framework, provides some interesting theoretical results that might help better understand offline methods such as CQL and IQL, and then under this framework, a new offline RL method called Sparse Q-learning (SQL) is proposed.

Empirical results from D4RL benchmark shows a consistent superior performance over competing algorithms. Authors also provided some other analysis on SQL and also compared to IQL on noisy data regime and CQL on small data regime (mixed datasets made from D4RL datasets).

**Summary Of The Review:**

Overall seems a solid paper, some interesting results and theoretical analysis. Overall writing is good.

My main concern:
- Additional technical details and whether other competing methods also had thorough hyperparameter tuning on each task should be provided, also the computation efficiency comparison, and final hyperparameters for each dataset would be good to have in the paper.
- I'm looking for a bit more analysis on your method, for example on the sparsity thing

I'm willing to increase my score if concerns are fully addressed and if anything is wrong in my comments please point it out.

=============== post rebuttal ===============
Sorry for the late reply, I thank the authors for their effort in addressing the concerns and providing the additional results. The new results seem to be more convincing. I have also checked other reviewers' comments and it seems most of the major concerns are addressed by the rebuttal. Based on that I increase my score to 8.

---

> ### Author Response · Authors · 2022-11-13
> **Response to Reviewer u1bC**
>
> We thank the reviewer for the thorough and detailed comments.
>
> >"for the methods you compare to, are they using the same hyperparameter for all tasks, or do they also search hyperparameter on each individual dataset and report the best one?"
>
> For the two strongest baselines, IQL and CQL, we initially used scores reported in their paper, except for CQL on AntMaze datasets, as we found the performance can be improved by carefully sweeping the hyperparameter $\texttt{min-q-weight}$ in $[0.5, 1, 2, 5, 10]$.
>
> As you suggested, for a fair comparision, we re-run IQL on all datasets and report the score of IQL by choosing the best score from $\tau$ in $[0.5, 0.6, 0.7, 0.8, 0.9, 0.99]$, using author-provided implementation (https://github.com/ikostrikov/implicit_q_learning). We also re-run CQL on all datasets and report the best score from $\texttt{min-q-weight}$ in $[0.5, 1, 2, 5, 10]$, using a popular PyTorch-version implementation (https://github.com/young-geng/CQL).
>
> We have updated the scores in Table 1 in our paper, we find that the reported scores in CQL and IQL papers are almost the highest one over the searched hyperparameters, SQL still outperforms both CQL and IQL, please refer to the updated paper for details.
>
> >"Similarly, in the case when you compare SQL to CQL and IQL on the noisy and small data regime settings, did you also do a thorough hyperparameter search for IQL and CQL?"
>
> On the noisy and small data regime settings, we did a thorough hyperparameter search for IQL and CQL, we reported the score of IQL by choosing the best score from $\tau$ in $[0.5, 0.6, 0.7, 0.8, 0.9]$, we reported the score of CQL by choosing the best score from $\texttt{min-q-weight}$ in $[0.5, 1, 2, 5, 10]$.
>
> >"Maybe I missed sth but have you provided empirical evidence and figures to demonstrate that your method has a more "sparse" policy than the other competitive methods?"
>
> Thanks for pointing out that, We want to clarify that having a "sparse" policy (only apply sparsity in the policy extraction) is not enough for good performance, we have tried using the policy extraction objective of SQL (i.e., equation (14)) to IQL in our preliminary experiments, but it doesn't get a better policy.
>
> The sparsity actually helps to learn a better value function (as we initially claimed in Section 4.4), we have added a whole section to elaborate how does sparsity benefit value learning in SQL, in both tabular setting and continuous action setting, please refer to Appendix B for details.
>
> >"What is the final hyperparameter you selected for each individual dataset? (You mentioned the best is selected for each dataset, would be good if you also report the actual numbers)"
>
> We list the per-dataset $\alpha$ of SQL in Table 7. SQL doesn't need to carefully select $\alpha$ as it is robust to a range of different $\alpha$. We can unify $\alpha$ to the following choices:
> - MuJoCo medium and medium-replay: $\alpha=2$
> - MuJoCo medium-expert: $\alpha=5$
> - AntMaze: $\alpha=0.5$ (except for antmaze-umaze-diverse, we set $\alpha=2$)
> - Kitchen: $\alpha=2$
>
> >"What about computation efficiency? How fast in wall-clock speed does your method compare to others?"
>
> We list the runtime of all algorithms in Table 9. SQL has almost the same wall-clock speed as IQL (20m), it can be expected because they only differ in the learning objective of $V$ and $\pi$. However, we want to mention that SQL has faster convergence compared to IQL.
>
> >"Hyperparameter sensitivity: for example can you provide some results on how performance changes with different alpha values?"
>
> We provide the sensitivity of $\alpha$ in SQL in Table 5 and Table 6, we provide the sensitivity of $\tau$ in IQL in Table 8. We found that SQL is not much sensitive to a range of different $\alpha$.
>
>
> >"Will your code be open sourced?"
>
> We will open-source the code and datasets, we have added a "Reproducibilty Statement" section to ensure that.

---

> ### Author Response · Authors · 2022-11-19
> **Follow up**
>
> Dear Reviewer u1bC,
>
> > "I'm willing to increase my score if concerns are fully addressed"
>
> We have included the technical details you mentioned and provided more analysis about the sparsity in the updated paper.
> Only a few hours are left in the discussion. **We would really appreciate it if you could tell us if your concerns are resolved**.
> Reviewer HMTv found that their concerns were addressed, and we have attempted to address all of your concerns.
> We would be more than happy to resolve any remaining questions in the time we have, and would like to engage in a discussion.
>
> Thanks!

---

> > ### Comment · Reviewer_u1bC · 2022-12-10
> > **Thank you for your rebuttal, I have increased my score**
> >
> > Sorry for the late reply, I thank the authors for their effort in addressing the concerns and providing the additional results. The new results seem to be more convincing. I have also checked other reviewers' comments and it seems most of the major concerns are addressed by the rebuttal. Based on that I increase my score to 8.

---

### Official Review · Reviewer_PvW6 · 2022-10-25

**Confidence:** 3
**Correctness:** 4
**Technical Novelty And Significance:** 4
**Empirical Novelty And Significance:** 3
**Recommendation:** 8

**Clarity, Quality, Novelty And Reproducibility:**

The paper is written well and clearly. It gives both new theoretical insights and a practical algorithm that achieve strong performance.

**Strength And Weaknesses:**

The paper is well-written and does a good job in explaining relevant concepts. The theoretical analysis gives interesting insights and shows also how prior algorithms (IQL and CQL) fit into this framework. Further relevant work is nicely summarized in the context of the paper and put in comparison to the proposed method.
The proposed method Sparse Q-learning (SQL) is derived from the theoretical framework under reasonable and well-explained assumptions.
SQL achieve SOTA performance on the D4RL benchmark and also strong results for the noisy and the small data regime.

Some typos:
p.2 "implicit value regularization" -> "implicit value regularization"
p.2 "IQL remains much close to the..." -> "IQL remains much closer to the..."
p.2 "...similarities to IQL However..." "...similarities to IQL. However..."
p.3 In the last equation defining the policy evaluation operator for V: There should be a plus instead of a minus
p.4 "... adds a entropy..." -> "... adds an entropy..."
p.5 the sentence " U∗(s) can be uniquely solved from the equation obtained by ..." appears twice in two consecutive paragraphs.
p.9 "This simulates the situation where the dataset is fewer and has limited state coverage near the target location because the data generation policies maybe not be satisfied and are more determined when they get closer to the target location"  what is meant here??

**Summary Of The Paper:**

The paper studies offline reinforcement learning by proposing the Implicit Value Regularization (IVR) framework. In IVR general regularizers are added to the policy learning objective. Algorithms from prior work are analyzed in this framework showing some of their weaknesses. From these theoretical insights, a practical algorithm is derived that achieves strong results on the D4RL benchmark.

**Summary Of The Review:**

This is an interesting paper both in terms of theory and practical results. Hence, I vote for acceptance.

---

> ### Author Response · Authors · 2022-11-13
> **Response to Reviewer PvW6**
>
> Thanks for the minor writing comments you posted, we have made several writing adjustments within the paper to reflect your concerns and suggestions, please see our revision.
>
> >"p.9 "This simulates the situation where the dataset is fewer and has limited state coverage near the target location because the data generation policies maybe not be satisfied and are more determined when they get closer to the target location" what is meant here??"
>
> We have revised this sentence to make it more clear. In Section 5.3, we want to simulate challenges one might encounter when using offline RL algorithms on real-world data. In real-world scenarios, the dataset size may be small or the dataset diversity of some states may be small. For example, in robotic manipulation tasks such as grasping, if the robot is not near the object, it can encounter diverse states by taking different actions and still pick up the object by the end; this is because unless the object breaks, actions taken by the robot are typically reversible, but when the robot grasps the object, its behavior should be more deterministic to ensure successful grasp without damaging or dropping the object.

---

### Author Response · Authors · 2022-11-13
**Revision Summary**

We thank all the reviewers for the detailed and constructive comments. We have revised the paper to address the concerns of the reviewers. The summary of changes in the updated version of the paper is as follows:

1. We re-run IQL and CQL on all datasets and report the best scores in Table 1.
2. We add the standard deviation of IQL and CQL in Table 1.
3. We list the per-dataset $\alpha$ of SQL in Table 7.
4. We provide the sensitivity of $\alpha$ in SQL in Table 5 and Table 6, and provide the sensitivity of $\tau$ in IQL in Table 8.
5. We use rliable library to validate the claim for SOTA results of SQL in Figure 7.
6. We list the runtime of all algorithms in Table 9.
7. We derive another practical new offline RL algorithm, EQL, by applying the reverse KL divergence in the IVR framework in Appendix A, we find EQL is an "implicit" version of AWR/AWAC while has a strong empirical performance.
8. We add a whole section to elaborate how sparsity benefits value learning in SQL, in both tabular setting and continuous action setting, in Appendix B.
9. We add a "Reproducibilty Statement" section to ensure reproducibilty.

We also want to highlight that the contributions of our paper is not only proposing a new effective offline RL algorithm, but more importantly, we propose a general Implicit Value Regularization framework, which gives deeper theoretical understanding of various existing offline RL methods and builds the bridge between behavior regularized and in-sample learning methods in offline RL. We believe that the proposed IVR framework also has the potential to scale to other settings, such as online RL and online/offline imitaiton learning.

---

### Decision · Program_Chairs · 2023-01-20

**Decision:**

Accept: notable-top-5%

**Justification For Why Not Higher Score:**

N/A

**Justification For Why Not Lower Score:**

The reviewers reach unanimous agreement that this paper presents both a general framework for offline RL with interesting theoretical insights and a practical algorithm that achieves SOTA performance.

**Metareview: Summary, Strengths And Weaknesses:**

This paper proposes the Implicit Value Regularization framework for offline reinforcement learning. This framework includes existing works such as the in-sample learning method, IQL, and value regularized method, CQL. The authors further propose a new practical algorithm, Sparse Q-learning that show SOTA performance on the D4RL benchmark.

Strengths:
- Theoretical analysis provides useful insight to show the weakness of existing algorithms
- A new practical algorithm that introduces sparsity in learning the value function, achieves better results and faster convergence.
- The authors also provide a section to analyze the sparsity characteristic of SQL in appendix B following the suggestion of a reviewer.

Weaknesses:
- The additional hyperparameter needs to be tuned per dataset, though the authors show the choice of hyperparameter is relatively robust.

The major concerns of the reviewer have been addressed after the rebuttal and all agree that this paper presents good contributions to offline RL.

**Note From Pc:**

if the above contains the word "oral" or "spotlight" please see: "oral" presentation means -> notable-top-5% and "spotlight" means -> notable-top-25%. As stated in our emails, we are disassociating presentation type from AC recommendations

**Summary Of Ac-Reviewer Meeting:**

N/A